# The Effect of *Lactiplantibacillus plantarum* BX62 Alone or in Combination with Chitosan on the Qualitative Characteristics of Fresh-Cut Apples during Cold Storage

**DOI:** 10.3390/microorganisms9112404

**Published:** 2021-11-22

**Authors:** Qian Zhao, Shihua Tang, Xiang Fang, Zhuo Wang, Yu Jiang, Xusheng Guo, Jianning Zhu, Ying Zhang

**Affiliations:** 1School of Public Health, Lanzhou University, Lanzhou 730000, China; zhaoq2020@lzu.edu.cn (Q.Z.); tangshh19@lzu.edu.cn (S.T.); fangx17@lzu.edu.cn (X.F.); zhwang2018@lzu.edu.cn (Z.W.); jiangy20@lzu.edu.cn (Y.J.); 2State Key Laboratory of Grassland and Agro-Ecosystems, School of Life Sciences, Lanzhou University, Lanzhou 730000, China; guoxsh07@lzu.edu.cn; 3Inspection Center, Gansu Medical Products Administration, Lanzhou 730070, China

**Keywords:** fresh-cut apples, *Lactiplantibacillus plantarum*, bio-preservative, chitosan

## Abstract

In order to explore whether beneficial lactic acid bacteria (LAB) could prolong the shelf life and improve the quality of fresh-cut apples, *Lactiplantibacillus plantarum* BX62, which was isolated from traditional fermented yak yogurt, and chitosan (CT), were applied to fresh-cut apples, subsequently stored at 4 °C. On days 0, 2, 4, 6, and 8, apple slices were taken for physicochemical, microbiological analysis, and sensory evaluation. The results showed that apple slices coated with *L. plantarum* BX62 (8 log CFU/mL) presented lower weight loss and browning rate, higher DPPH scavenging capacity, and achieved more effective inhibition of polyphenol oxidase (PPO) and peroxidase (POD) activities compared to the control samples. The application of CT alone or in combination with *L. plantarum* BX62 resulted in a significant reduction in aerobic mesophilic bacteria (AMB), aerobic psychrophilic bacterial (APB), yeast and molds (YAMs) counts (2.31 log CFU/g for AMB, 2.55 for APB, and 1.58 for YAMs). In addition, *L. plantarum* BX62 coated apples showed acceptable sensory properties in terms of color, flavor, taste, texture, and overall visual quality during 8 d of storage. On this basis, *L. plantarum* BX62 could be used as an excellent bio-preservative to extend the shelf life and improve the quality of fresh-cut apples.

## 1. Introduction

Fresh-cut fruits, also nominated as minimally processed fruits, are fresh, nutritious, convenient, pollution-free, and 100% edible [1]. In recent years, consumers’ requirement for fresh-cut fruits has increased, leading to a dramatic expansion in market demand [2]. The shelf life of fresh fruits after cutting is very short, usually 5 to 7 days under cold temperature, because peeling and cutting damages the cell structure around the injuring point, leading to browning and nutrient loss, and thus shortening the fruit’s shelf life [3]. In addition, when fruits are processed, microorganisms may be transferred from the surface to edible parts and become potential vectors of major foodborne pathogens, such as *Salmonella*, *E. coli*, and *L. monocytogenes* [4]. Therefore, it is challenging to prolong the shelf life and ensure the safety of fresh-cut fruits, but it can be achieved by physical, chemical, or biological methods or a combination of two or more of the above methods [5].

Edible coatings and films are a feasible, promising, and innovative fresh-keeping method in maintaining the quality as well as the shelf life of fruits, which can be eaten together without further processing [3]. In foreign countries, edible coating materials enriched with anti-browning or antibacterial substances have long been used in fruits and vegetables [6]. However, the fresh-cut products industry in China started relatively late and currently mainly adopts physical and chemical preservation technologies or their combination. Most physical methods are high-cost and difficult to implement in large scale production [7]; whereas chemical methods may be easier to carry out, most can generate harmful residues, which may endanger the health of consumers, and the extensive use of chemical preservatives may also cause severe environmental pollution [8]. Therefore, looking for safe and eco-friendly preservation technology has become a critical issue from all walks of life. Biological preservation has the advantages of non-toxicity, no residues, no secondary pollution, and no drug resistance; it is an effective way to achieve pollution-free preservation of fruits, and is also one of the development directions of new biological preservatives [9].

Chitosan (CT) is a natural macromolecular polysaccharide that has been extensively applied as an edible coating for its film-forming characteristics and is also valued as a functional polymer that is highly attractive for various industries due to its good bio-compatibility, bio-degradability, and excellent antimicrobial capacity [10]. There have been many reports about the application of chitosan-based edible coatings on fruit [11,12,13]. Saleem et al. [13] reported that CT (1% *w*/*v*) containing ascorbic acid (1% *w*/*v*) can reduce weight loss and inhibit softening of strawberries during storage, and their sensorial properties were not affected significantly. With further development of edible coatings and films, the edible coating can be used as a carrier of probiotics to exert antibacterial and antioxidant effects, to prolong the shelf life of perishable fruits.

Lactic acid bacteria (LAB) have long been consumed in dairy products, and many are classified as “generally recognized as safe” (GRAS) microorganisms because they are nonpathogenic, suitable for technological and industrial processes, and have the ability to produce antimicrobial substances [14]. They are considered to be one of the most important groups of bacteria for their positive roles in the food industry, contributing to ensuring the quality of fruits, vegetables, and other food products through antibacterial or fermentation activity [15]. They can secrete lactic acid, hydrogen peroxide, and peptides to inhibit or block the adhesion of epithelial cells and the contamination of intestinal pathogens. In order to control the microbial contamination of fresh-cut products, people have studied different technologies, one of which is the use of LAB as well as their metabolites for biological preservation [4]. *L**. rhamnosus* [9], *L. casei* [14], *L. plantarum* [16], and *L*. *pentosus* [17] have been reportedly used in the preservation of fresh and minimally processed fruits and vegetables, and these LAB could enhance the quality and prolong the shelf life of products by inhibiting the growth of pathogenic microorganisms. Sorrentino et al. [18] found the use of *L. plantarum* alone or in combination with a sterile gel could extend the shelf life of fresh truffles by avoiding spoilage due to molds, as well as inhibiting the growth of *Penicillium digitatum* DSM 2750, a green mold involved in the spoilage of truffles. Similar results were found by Siroli et al. [19], who treated apple slices with *L. paracasei* M3B6 and *L. plantarum* CIT3; both strains can ensure the final product’s safety and extend the shelf life. In addition, *L. plantarum* CIT3 can prolong the shelf life of fresh-cut apples to 9 days when used alone and 16 days when combined with natural antimicrobials.

Traditional fermented yak yogurt on the Qinghai-Tibet Plateau is rich in LAB resources*, Lactobacillus delbrueckii* subsp. *bulgaricus* Strain F17 and *Leuconostoc lactis* Strain H52 were isolated by our team in the early stage. According to previous research, both F17 and H52 have exhibited excellent probiotic properties, which can significantly reduce the weight loss rate and decay rate of fresh grapes and strawberries and inhibit the growth and colonization of microorganisms, such as aerobic mesophilic bacteria, mold, yeast, and *E. coli* [20,21]. *L. plantarum* BX62 possesses favorable antioxidant activities and can be isolated from traditional fermented yak milk. Chen et al. [22] reported that the scavenging rate of *L. plantarum* BX62 suspension (10 log CFU/mL) on DPPH radical was 75.17%, and the scavenging rates of hydroxyl radical and superoxide anion radical were 81.13% and 65.85%, respectively—significantly higher than the standard strain *Lactobacillus rhamnosus* GG (LGG; ATCC53103) (Purchased from China General Microbiological Culture Collection Center, CGMCC) [23,24]. Therefore, *L. plantarum* BX62 was chosen to carry out the fresh preservation experiment on fresh-cut apples.

Since there is a growing interest in the study of lactic acid bacteria as bio-preservatives to ensure food safety and prolong the shelf-life of fresh and minimally processed fruits and vegetables, exploring the effects of *L. plantarum* BX62, with or without chitosan, on the quality and shelf-life of fresh-cut apples could be highly encouraging. The combination of the antimicrobial and barrier properties of chitosan and the high antioxidant properties of BX62 could yield a more promising method to prolong the shelf-life and improve the quality of fresh-cut apples, which will be beneficial to producers and the wider food industry.

## 2. Materials and Methods

### 2.1. Materials

#### 2.1.1. Apple Samples

“Red Fuji” apples originating from Jingning County, Gansu Province, were purchased from local fruit supermarkets in Lanzhou. Eighty apples of uniform shape, color, size, and similar maturity, and without mechanical and pathological injuries were selected and stored in a refrigerator at 4 ± 1 °C before processing [25].

#### 2.1.2. Preparation of Coating Solution

The *L. plantarum* BX62 were provided by the laboratory of the “Probiotics and Biological Feed Research Center” of the School of Life Sciences, Lanzhou University. Before the test, cultures of *L. plantarum* BX62 were thawed at room temperature and then inoculated on Man-Rogosa-Sharpe (MRS, Qingdao Hope Bio-Technology Co., Ltd., Qingdao, China) solid media with an inoculation loop. After culturing at 37 °C for 24 h, a single colony was picked and inoculated into tubes containing MRS broth. The tubes were kept at 37 °C for 48 h and then centrifuged at 5000 r/min for 15 min at 4 °C. The supernatant was discarded, and the pellets were collected and washed 3 times with sterile normal saline at 5000 r/min for 15 min at 4 °C. Finally, the cell concentration was adjusted to 1 × 10^8^ CFU/mL.

The chitosan solution was prepared according to Martínez et al. [26]. Chitosan (degree of deacetylation ≥ 95%, Beijing Coolaber Technology Co., Ltd., Beijing, China) was added to the 1.0% (*v*/*v*) acetic acid solution to obtain a final concentration of 1.0% (*w*/*v*), stirred until completely dissolved, and then the pH value was adjusted to 4.0.

The *L. plantarum* BX62 bacterial suspension was mixed with chitosan solution to prepare a Chitosan-*L. plantarum* BX62 Solution (CT + BX62), The concentration of chitosan was 1.0%, and that of *L. plantarum* BX62 was 1×10^8^ CFU/mL, and then adjusted to pH 4.0.

### 2.2. Application of the Coatings

Apples were placed at room temperature for 2 h before treatment, and then washed with 1% (*v*/*v*) sodium hypochlorite solution for 2 min and rinsed twice with sterile distilled water to remove residual chlorine. Rinsed apples were wiped with absorbent paper to remove excess water and aseptically peeled, cored, and cut using a sterile fruit knife to acquire 8 pieces. Each piece was distributed to a different group [27].

A total of 640 apple pieces were used in the experiment and were randomly assigned to four groups. Then the coatings were applied for four groups, which included T1 = without any treatment, T2 = CT solution, T3 = *L. plantarum* BX62, T4 = CT + BX62 (1% CT + 1 × 10^8^ CFU/mL BX62). Apple pieces were immersed in the coating solutions for 30 s and then dried at room temperature. Treated apple pieces were packed in plastic boxes and sealed with breathable PE plastic wrap. The boxes were put in a refrigerator at 4 °C for 8 days after weighing. The physicochemical properties, microbial counts, and sensory evaluation were performed on days 0, 2, 4, 6, and 8. All experiments were performed in triplicate.

### 2.3. Determination of Weight Loss Rate

The weights of the apple slices were measured every other day for each group and recorded as W_0_, W_2_, W_4_, W_6_, and W_8_. The weight loss rate (WLR) can be calculated by the following formula:Weight loss rate (%) = (W_0_ − W_t_)/W_0_ × 100%(1)
where W_0_ denotes the weight of apple pieces on day 0, W_t_ denotes the weight of apple pieces on days 2, 4, 6, and 8.

### 2.4. Browning Rate Assessment

The browning rate (BR) of fresh-cut apple samples was subjectively based on a numeral scoring index using a scale of 0–4, where 0 = no browning, 1 = slight browning (browning area less than 5%), 2 = moderate browning (reaches 5–20%), 3 = moderately severe browning (reaches 20–50%), and 4 = severe browning (more than 50%) for individual apple slices. The BR was calculated as follows: browning rate (%) = ((% apple slices with slight browning × 1) + (% apple slices with moderate browning × 2) + (% apple slices with moderately severe browning × 3) + (% apple slices with extreme browning × 4))/5 [28].

### 2.5. Measurement of TA and SSC

Fresh fresh-cut apple juice was filtered and used for the measurement of titratable acidity (TA) and total soluble solids content (SSC). The TA was measured according to Youssef et al. [29], modified to the combination of potentiometric titration and indicator titration. Five milliliters of apple juice were diluted with equal amounts of distilled water plus the addition of 2 drops of phenolphthalein as an indicator and titrated with 0.1 mol/L NaOH standard solution, and the endpoint was pH = 8.10–8.20. The result was expressed as % tartaric acid. The filtered apple juice was centrifuged at 10,000× *g*/min for 10 min at 4 °C. After centrifugation, 100 μL supernatant was taken to measure the total SSC with a hand-held refractometer (RHB-18ATC, Shanghai, China). The results were expressed as °Brix [30].

### 2.6. Determination of TPC

The determination of total phenolic content (TPC) was compiled by the Folin-Ciocalteu method of Lin et al. [28]. Apple samples (5 g) were ground to homogenates in methanol (5 mL) solution, and then extracted 3 times by ultrasound (15 min each time), and centrifuged at 10,000 r/min for 10 min, the supernatant was collected and diluted to 25 mL to prepare the sample solution. The reaction solution consisted of 400 μL sample solution, 600 μL distilled water, 1 mL Folin-Ciocalteu’s phenol reagent (Beijing Solarbio Technology Co., Ltd., Beijing, China), and 5 mL sodium carbonate (*w*/*v*, 20%) solution. The absorbance of the mixture was carried out after 30 min of reaction by an ultraviolet spectrophotometer (U-2910, Hitachi, Japan) at 725 nm, and the results were expressed as mg equivalents of gallic acid (GAE) per 100 g of fresh weight (FW) for each treatment.

### 2.7. Assay of Total Antioxidant Activity

The total antioxidant capacity of fresh-cut apples was measured by 1,1-diphenyl-2-picrylhydrazyl (DPPH) radical scavenging capacity in accordance with the method of Moreira et al. [31]. The DPPH analysis provides an estimate of the total antioxidant ability, not aimed at special antioxidant compounds of the sample. A 3.9 mL DPPH solution (0.025 g/L) in methanol was mixed with 0.1 mL of apple supernatant. The mixture was shaken vigorously and kept in darkness for 30 min at 25 °C. Then absorbances were read at 515 nm taking methanol without DPPH as the control. The DPPH scavenging activity was calculated as follows:DPPH scavenging capacity (%) = (Ac − As)/Ac × 100(2)
where Ac denotes the absorbance of the control at 515 nm and As the sample.

### 2.8. Enzymatic Assays

Apple pieces (5 g) were ground in 9 mL of phosphate-buffered saline (PBS, 0.1 mol/L, pH 7.0) and 1 mL of polyvinylpyrrolidone (PVPP) (0.2%, *w*/*v*) in an ice bath for 5 min, and then the supernatant was collected for enzyme activity determination after centrifugation at 10,000 rpm for 15 min at 4 °C. The determination of polyphenol oxidase (PPO) and peroxidase (POD) activities was based on the method of Lin et al. [32].

The activity of PPO and POD can be obtained by the following formulas:PPO activity (U/g FW·min) = (ΔA420 × V)/(0.01t × vs. × m)(3)
POD activity (U/g FW·min) = (ΔA470 × V)/(0.01t × vs. × m)(4)

In the formula, ΔA420/ΔA470 is the absorbance change at 420 nm/470 nm of the mixtures. V is the total volume of sample extract, vs. is the volume of sample extract used, m is the mass of the sample, and t is reaction time.

### 2.9. Microbiological Analysis

Aerobic mesophilic bacteria (AMB), aerobic psychrophilic bacterial (APB), yeast and molds (YAMs), and LAB on fresh-cut apples were periodically evaluated as previously described by Graca et al. [33]. A 25-g sample of apple slices was immersed in 225 mL of sterile physiological saline solution and then shaken at 60 rpm/min for 35 min at 25 °C by a constant temperature culture oscillator (SPH-2102, Shanghai Shiping Experimental Equipment Co., Ltd., Shanghai, China). After that, gradient dilutions were poured into plate count agar (PCA) to determine AMB and APB, potato dextrose agar (PDA) to count YAMs, and MRS solid media to enumerate LAB. PCA, PDA, and MRS plates were incubated, respectively at 37 °C for 1 to 2 d (for AMB), 7 °C for 8 to 10 d (for APB), 25 °C for 3 to 5 d (for YAMs), and 37 °C for 2 to 3 d (for LAB). After incubation, the plates containing between 30 to 300 colonies were recorded. Results were expressed as log colony forming units per g of fruit (log CFU g^−1^). All tests were run in triplicate.

### 2.10. Sensory Evaluation

Different sensory parameters (flavor, color, odor, texture, overall visual quality (QVQ)) of apple pieces were carried out by a panel of trained judges (*n* = 5) on days 0, 2, 4, 6, 8. Four pieces per sample were randomly served and placed at room temperature for one hour before sensory analysis. Each judge was asked to evaluate apple slices independently, with a score of 0–10, to quantify the intensity of the attributes following the methodology described by Alvarez et al. [34], with a slight modification. Scores for flavor from 0 (intense unpleasant flavor) to 10 (typical-no odd flavor); color from 0 (high presence of browning) to 10 (absence-no browning); odor from 0 (strong off-odors) to 10 (fresh-like odor); texture from 0 (very soft) to 10 (crispy); and OVQ from 0 (highly deteriorated appearance) to 10 (fresh appearance). The limit of acceptance was 6.0 (60% of the scale), which indicates that a score below 6.0 for any evaluation parameter is considered to be the end of shelf-life. The final results were the average score of all team members.

### 2.11. Statistical Analysis

All data were expressed as the means ± standard deviations of three independent experiments in each group. SPSS 26.0 software and the Tukey test were used to perform a one-way analysis of variance and to determine statistically significant differences between the treatments at each storage time. Pearson correlation analysis was used to analyze the correlation between different indicators. The significance level of all the above statistical tests was defined at *p* < 0.05.

## 3. Results

### 3.1. Weight Loss

As shown in Figure 1, the weight loss rate of apple pieces in all groups increased gradually with the extension of storage time and was more pronounced for the control samples from day 2 to day 8 (*p* < 0.05). The possible reason for the more prominent weight loss in the control group is that the semi-permeable barrier formed by chitosan and the adhesion of *L. plantarum* BX62 can act against water loss. However, statistical analysis did not show a significant difference in WLR among CT, BX62, and CT + BX62 coatings (*p* > 0.05), the WLR of the most serious control group was not more than 0.25%, which indicated that low-temperature storage and breathable PE film packaging could reduce the transpiration and respiration of fresh-cut apples.

### 3.2. Browning Rate

Browning rates of fresh-cut apples subjected to the different coating formulations were assessed during the storage period, as shown in Figure 2. The browning rate gradually increased with the extension of storage time between every treatment, and the apple pieces treated with CT presented obviously higher browning rates than that of the other three groups (*p* < 0.05), while the apples treated with BX62 presented significantly lower browning rates than the other three groups (*p* < 0.05), BX62 coatings markedly reduced browning rate to about 68.75% of CT group after eight days of storage. However, the browning rate of the control and the CT + BX62 groups were not statistically different (*p* > 0.05).

### 3.3. TA and SSC

It can be seen from Figure 3a that during storage, the TA content of fresh-cut apples in the control group and BX62 group showed a slight increase during the first two days of storage and a slow decrease thereafter. Meanwhile, the TA content of fresh-cut apples in the CT group and CT + BX62 group showed a sharp decrease on day two and then a significant increase. The total SSC of fresh-cut apples in all groups showed a stable downward trend from day 0 to day 8, and there was no significant difference among groups (*p* > 0.05) (Figure 3b).

### 3.4. Total Phenolic Content

The total phenolic content of fresh-cut apples in each group showed a tendency to increase at the early stage of storage and then decreased slowly at 2–8 days of storage (Figure 4). Notably, the TPC under the BX62 treatment was the highest among all treatments, while the CT treatment was the lowest (*p* < 0.05). Over the entire storage period, the TPC of apples coated with CT + BX62 was in the intermediate level compared with the CT and BX62 group (*p* < 0.05).

### 3.5. Total Antioxidant Activity

The scavenging capacity for DPPH reflects the antioxidant capacity of fresh-cut apples during storage. It can be seen from Figure 5, that the DPPH scavenging capacity of fresh-cut apples markedly declined in all four groups during storage. Meanwhile, the DPPH scavenging capacity of the BX62 group showed a higher level compared with the CT + BX62 group, suggesting that the antioxidant activity of BX62 is superior to the combination of CT and BX62. In addition, the DPPH radical scavenging rate of the CT group was significantly lower than that of the control group and CT + BX62 group (*p* < 0.05).

### 3.6. Enzyme Activities

The evolution of polyphenol oxidase (PPO) and peroxidase (POD) activities is shown in Figure 6. The activity of PPO in all groups showed a stable, increasing trend during 0–4 days and then gradually decreased with the prolonging of storage time (Figure 6a). By contrast, the POD activity of fresh-cut apples increased throughout the storage period in all four groups. During storage, the activities of PPO and POD in the BX62 group were significantly lower than those in the other groups (*p* < 0.05). In addition, compared with the CT group, the control and CT + BX62 groups showed lower PPO and POD activities (*p* < 0.05) (Figure 6b). However, there were no statistical differences in PPO and POD activities between the control and the CT + BX62 treatment (*p* > 0.05).

### 3.7. Microbiological Analysis

The aerobic mesophilic bacteria (AMB), aerobic psychrophilic bacteria (APB), yeast and molds (YAMs), and lactic acid bacteria (LAB) counts for different apple treatments throughout storage are presented in Table 1. During storage, the number of AMB for the CT + BX62 group was decreased gradually from day 2 to 8 and was lower than that in the control group (*p* < 0.05), in addition, AMB was not detected in CT-treated samples from day 2 to day 8. Studies on the viability of *L. plantarum BX62* in BX62 and CT + BX62 coating treatments during storage of apple pieces were carried out; the obtained results showed that there were no significant differences in the initial *L. plantarum* BX62 counts between treatments (BX62 and CT + BX62) with values in the range of 8.98–8.94 CFU/g. However, from the second day to the end of the storage, viable *L. plantarum* BX62 population in fresh-cut apples without CT presented LAB counts significantly (*p* < 0.05) higher. The number of APB on the fresh-cut apple surface treated by CT and CT + BX62 was 1 log (CFU/g) from day 2 to day 8. In addition, the number of APB colonies in the control group was higher than that of the BX62 treatment group from day 2 to day 6 (*p* < 0.05). Regarding YAMs, coating with CT and CT + BX62 reduced the initial counts of the samples; similarly, BX62 has no inhibitory effect on YAMs throughout the whole storage period. However, it is notable that the AMB, APB, and YAM growth was maintained below 6.0 log (CFU/g). We can conclude that the combination of CT and BX62 can achieve better antibacterial effects without considering browning.

### 3.8. Sensory Evaluation

The results of sensory characteristics of fresh-cut apples treated with *L. plantarum* BX62 alone or incorporated with chitosan are shown in Figure 7. Initially, chitosan treatment resulted in a slight color change of apple slices, while BX62 treatment had no effect on sensory properties when compared with the control group. The overall visual quality (OVQ), color, and texture of all treated samples decreased with the extension of storage time. It is highlighted that the OVQ, texture, and color of apple slices treated with BX62 decreased the least, and the flavor and odor remained relatively stable. While the flavor and odor scores of other groups decreased gradually from day 0 to day 8. The highest acceptability of fresh-cut apples treated with *L. plantarum* BX62 was found on day eight with a significant difference (*p* < 0.05), and the lowest acceptability was observed in C-treated apple slices (*p* < 0.05), the scores were lower than the established limit (6.0 points of scale). However, there was no difference in sensory evaluation between the control group and the CT + BX62 group during the cold storage.

### 3.9. Pearson Correlation Analysis

The physicochemical and microbiological information obtained from the experiment were analyzed by Pearson correlation analysis to explore the correlations between them; only the indicators of significant correlation are listed in Table 2. The counts of APB and YAMs on the surface of fresh-cut apples in the control group were positively correlated with the weight loss rate, browning rate, and activity of POD but negatively correlated with the DPPH scavenging rate and SSC. In addition, in the CT group, the counts of AMB, APB, and YAMs showed a significant positive correlation with SSC and a negative correlation with the activity of POD. Moreover, the counts of *L. plantarum* BX62 in BX62 and CT + BX62 treated apple pieces exhibited a significant negative correlation with weight loss rate, browning rate, and activity of POD, while it was positively correlated with SSC and DPPH scavenging rate. It is noteworthy that there was no significant correlation between the number of microbial communities (AMB, APB, YAMs, and LAB) and TA, TPC, and the activity of PPO in four groups (Appendix A).

## 4. Discussion

Traditional fermented yak yogurt on Qinghai Tibet Plateau is rich in lactic acid bacteria (LAB) germplasm resources, which has been widely used in various production environments, but it is rarely used in fruit and vegetable preservation; thus, we studied the application of *L. plantarum* BX62 with high antioxidant activity in fresh-cut apples.

Color and appearance are considered important indicators of the freshness of fruits. Both higher weight loss rate and higher browning rate may cause the degradation of fresh-cut apples’ sensory acceptance and market value [35]. The weight loss rate of fresh-cut apples in the three treatment groups was significantly lower than that in the control group. The possible reason for lower weight loss in the treated groups than in the controls is that the semi-permeable membrane formed by chitosan can inhibit the transpiration and respiration of fruits. Moreover, *L. plantarum* BX62 can reduce the activity of cell respiration-related enzymes, thereby reducing respiration, which subsequently contributes to maintaining the moisture of fresh-cut apples and thus prevents the loss of weight [1]. The results led to similar conclusions where grapes or strawberries were treated with the supernatants of *L. delbrueckii* subsp. *bulgaricus* Strain F17 and *Leuconostoc lactis* Strain H52, showing lower weight loss and decay rates [20,21].

Kumar et al. [36] treated fresh cut “Royal Delicious” apples with edible coatings and anti-browning agents, found the use of edible coatings, both separately and in combination with anti-browning agents, to be a very promising approach to ensure the quality of fresh-cut apples [37,38]. It was reported that the process of peeling and slicing could cause tissue damage and induce enzymatic and non-enzymatic reactions, resulting in oxidative browning of fresh-cut apples [39]. The browning rate of fresh-cut apples in the BX62 treatment group was the lowest, which reflected the good antioxidant capacity of BX62, and was consistent with the results of Chen et al. [24]; however, the mechanism of antioxidants needs to be further elucidated. However, the browning rate of fresh-cut apples under the CT treatment and the CT + BX62 treatment was higher, which did not achieve the expected effect, but may be related to the pH and concentration of chitosan solution and the crosslinking properties of chitosan and BX62.

TA and SSC are the most principally accepted quality parameters regarding the maturity of apples. The TA in fresh-cut apples is mainly organic acid, which can be used as a substrate for respiration and also serves as an important index to evaluate the sensory characteristics of fruits [40] and is mainly affected by water loss, respiration intensity, coating thickness, and microbial acid production [41]. Accordingly, the complex phenomenon caused by TA may be related to the decrease of organic acids, which is due to the increase of respiration rate after peeling and cutting [42]. However, the SSC of fresh-cut apples in each treatment group was higher than that in the control group. The possible reason was that different treatments, such as chitosan, weakened the metabolic activity of fresh-cut apples and reduced the consumption of soluble solids. Surprisingly, there was no correlation between TA and SSC, which changed in the opposite direction during apple ripening and decay (Appendix A).

Polyphenols in fresh-cut apples are their secondary metabolites, which play an important role in maintaining the nutritional and sensory quality of fruits and vegetables, which reflects the antioxidant capacity of fruits [43]. A plausible reason for the increase of total phenolic content at the early storage time may be that the cutting operation promoted the accumulation of bioactive substances, which thus stimulated the generation of polyphenols. The decrease of total phenolic content during 2–8 days might be due to polyphenols, as the substrate of phenolic enzyme oxidation browning reaction, were gradually consumed with the extension of storage period [44]. The apples coated with chitosan showed the lowest levels of phenols; the results were inconsistent with the results reported by Zhang et al. [45], which may be related to the protective barrier formed by chitosan coatings on apple surfaces that could inhibit the loss of phenolics.

The scavenging capacity for DPPH reflects the antioxidant capacity of fresh-cut apples during storage, which is determined by the type and amount of bioactive substances and provides an estimate of the overall antioxidant capacity of the sample as it is not specific for any appointed antioxidant compound [26]. The DPPH scavenging rate of the BX62 group was the highest during storage, which may be contributed to by the antioxidants produced by BX62.

Enzymatic browning of fruits and vegetables is mainly caused by mechanical and physical damage during postharvest processing and storage, which is mainly driven by polyphenol oxidase (PPO), an intracellular o-diphenol oxidase widely distributed in higher plants and fungi [46]. PPO is the main enzyme involved in the browning process of fresh-cut apples as they can form quinone compounds by catalyzing the phenolic substances [47]. PPO activity was significantly increased in the four groups during the storage period of 0–4 days. This may be due to the cutting operation stimulating the defense reaction, induced expression of PPO gene, and increased activity of PPO [30]. A possible reason for the decrease since day four is that the enzyme-substrate decreased with the prolongation of storage time, and the accumulation of browning products inhibited the activity of PPO to a certain extent [28]. POD is a key indicator in the stimulation of the enzymatic defense system, and this activity increases in the storage time [48]. POD can coordinate with other enzymes of fresh-cut apples to remove the reactive oxygen species (ROS) and maintain the ROS of fresh-cut apples in normal dynamic balance so as to improve the stress resistance of fresh-cut apples [49]. The POD activity of fresh-cut apples treated with BX62 was lowest during the whole storage period; it may be that BX62, with high antioxidant activity can eliminate the active oxygen-free radicals produced by cutting.

Pathogenic microorganisms in fresh-cut products are closely related to consumer health because these products generally do not receive any further treatment before consumption after entering the market [17]. Zhang et al. [50] found that minimally processed apples were more frequently contaminated by *L. monocytogenes* compared with other fresh-cut fruits. Microbial safety is the most critical aspect of food hygiene [42]. The peeling and cutting process can cause damage to cells around the injured points and lead to the leakage of cellular content, which is favorable to microbial growth, as these cell contents are rich in water, carbohydrates, vitamins, and other nutrients [51]. Our study underlined that the appearance of CT significantly affects the dynamic populations of *L. plantarum* BX62; it is supposed that CT is disadvantageous to the viability of lactic acid bacteria. The decrease of LAB populations in the BX62 treatment group during storage may be related to the adverse living conditions, especially aerobic conditions, and the competitive inhibition of pathogenic bacteria, such as yeasts and molds and other microorganisms. In addition, chitosan has excellent antibacterial properties and can inhibit the survival of AMB, APB, and YAMs, maintained below 6.0 log (CFU/g), which is the maximum value required by law [49]. Solís-Contreras et al. [52] reported the antimicrobial ability of three bioactive coatings on apple cubes; the chitosan-based coating is by far the best in delaying the growth of mesophilic bacteria, yeast, and molds. A study performed by Peralta-Ruiz et al. [53] showed that chitosan (2%, *w*/*v*) enriched with *Ruta graveolens* essential oil (0.5%, *v*/*v*) coatings can significantly reduce the counts of AMB and molds without affecting sensory acceptability.

Sensory evaluation of minimally processed products is crucial because it indicates the acceptability of the final consumer of the fruit. Fresh-cut apples coated with BX62 presented better flavor and odor characteristics (Figure 7) during the storage period, which may be related to the diversified metabolites produced by lactic acid bacteria [34]. Similar results were obtained by Nematollahi et al. [54], who reported that *L. casei* TD4-treated cherry juice showed better taste, odor, and overall acceptance than the control samples. Bambace et al. [34] added the probiotic *L. rhamnosus* CECT 8361 to edible coating containing prebiotics to treat fresh blueberries, and obtained better flavor and odor characteristics. Meanwhile, Khodaei et al. [55] found that the color, flavor, taste, and texture scores of strawberries coated with *L. plantarum* PTCC 1058 were higher than the minimum acceptable limit during 10 d of storage. In terms of fresh-cut apples treated with CT, the scores of sensory parameters were lower than those of the control group during storage. Consistent with our results, Solís-Contreras et al. [52] reported that 1.0% of chitosan (*w*/*v*) could cause off-odor in minimally processed apples.

However, when comparing our results to those of previous studies, it must be pointed out that the combination of chitosan and *L. plantarum* BX62 did not achieve optimal antibacterial and antioxidant effects; it may be related to the oxygen-sensitivity of lactobacillus and unfavorable acidic conditions. Clearly, chitosan has failed to play a positive role in the above issues. Results suggested that efforts should be made to counteract oxygen stress and to improve the viability of lactobacillus under aerobic conditions. The addition of prebiotics and natural antioxidants may be one of the effective ways to solve this problem. Yan et al. [56] found that persistent antioxidant juice of ‘Changchong’ pear is effective in improving the growth and viability of *L. plantarum* strain ST-III and other representative strains under aerobic and anaerobic conditions. In addition, metabolites of lactic acid bacteria such as bacteriocin, short-chain fatty acids (SCFA), and extracellular polysaccharides (EPS) [57] were reported to possess powerful antimicrobial or antifungal activities. Adding bacteriocin directly to fresh-cut fruits, or using edible coatings and films, may be another feasible method. Barbosa et al. [58] have demonstrated that nisin-incorporated cellulose films have strong antibacterial effects on *S. aureus*, *L. monocytogenes*, *L. acidophilus,* and *Bacillus cereus* without affecting the physicochemical properties of mangoes.

## 5. Conclusions

Results presented in this article demonstrated that the application of *L. plantarum* BX62 had the effectiveness of controlling browning, and chitosan had the effectiveness of reducing microbial colony counts. Fresh-cut apples treated with CT, BX62, and CT + BX62 all significantly decreased the weight loss rate compared with the control groups. *L. plantarum* BX62 coating beneficially enhanced the DPPH scavenging capacity, reduced total phenolic content and inhibited the activity of PPO and POD, and maintained acceptable sensory properties in terms of color, flavor, taste, texture, and overall visual quality during eight days of storage. However, CT and CT + BX62 solution failed to control the browning of fresh-cut apples; thus, the condition needs to be further carefully optimized. In addition, CT and CT + BX62 treatment significantly inhibited AMB, APB, YAMs, which indicates that CT exhibits good antibacterial effects. Moreover, the Pearson correlation analysis revealed that the antioxidant capacity of BX62 was highly correlated with DPPH scavenging rate and PPO and POD activity, which indicates that *L. plantarum* BX62 was an excellent natural antioxidant to delay the oxidative browning of fresh-cut apples. However, the antioxidant mechanism of *L. plantarum* BX62 needs to be further elucidated, and the compounding conditions of chitosan and BX62 need to be further explored to achieve better antibacterial and antioxidant effects. Future researchers are also required to focus on the impact of probiotics on volatile flavor compounds and the role of probiotics in vegetables and fruits, to drive the research from laboratory to market.

## Figures and Tables

**Figure 1 microorganisms-09-02404-f001:**
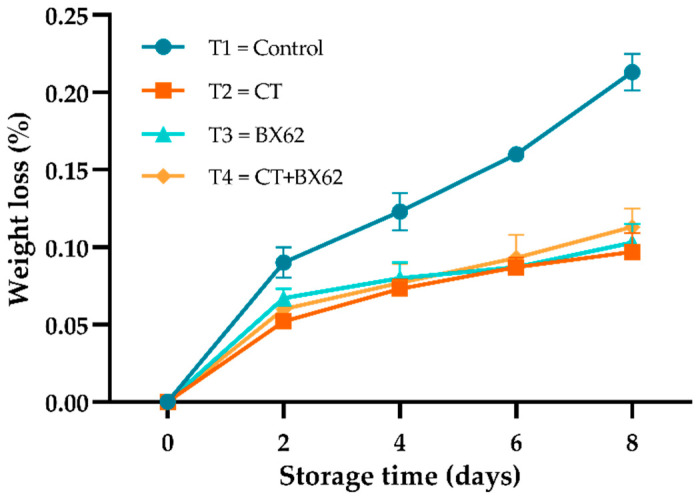
Changes in weight loss rate for the control apples (●), chitosan-treated (CT) apples (■), *L. plantarum* BX62-treated (BX62) apples (▲), and chitosan in combination with *L. plantarum* BX62-treated (CT + BX62) apples (◆) after being stored at 4 °C for 8 days. Each data point is the mean of three replicate samples.

**Figure 2 microorganisms-09-02404-f002:**
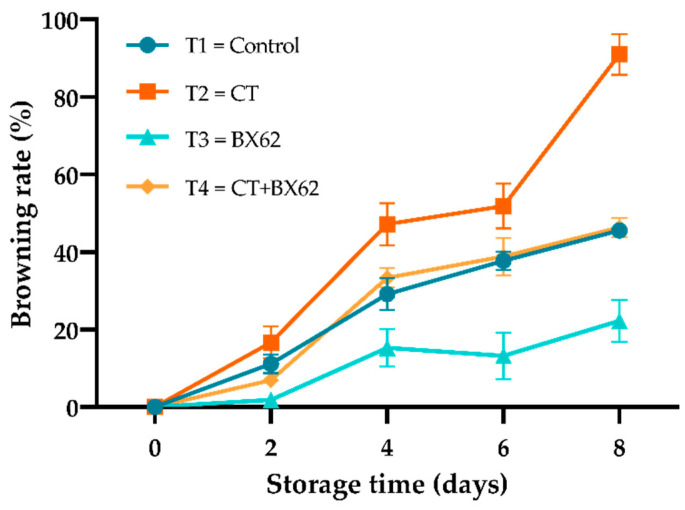
Changes in browning rate for the control apples (●), chitosan-treated apples (■), *L. plantarum* BX62-treated apples (▲), and chitosan in combination with *L. plantarum* BX62-treated apples (◆) after being stored at 4 °C for 8 days. Each data point is the mean of three replicate samples.

**Figure 3 microorganisms-09-02404-f003:**
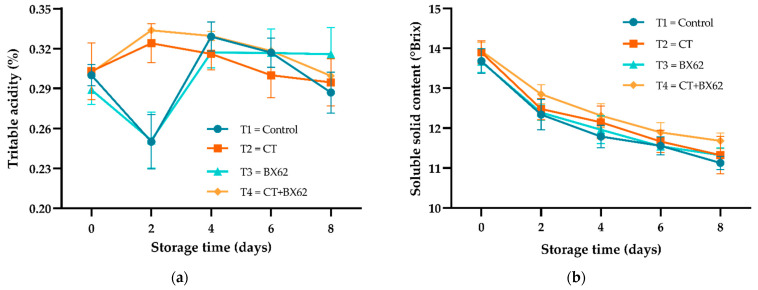
Changes in TA (**a**) and SSC (**b**) for the control apples (●), chitosan-treated apples (■), *L. plantarum* BX62-treated apples (▲), and chitosan in combination with *L*. *plantarum* BX62-treated apples (◆) after being stored at 4 °C for 8 days. Each data point is the mean of three replicate samples.

**Figure 4 microorganisms-09-02404-f004:**
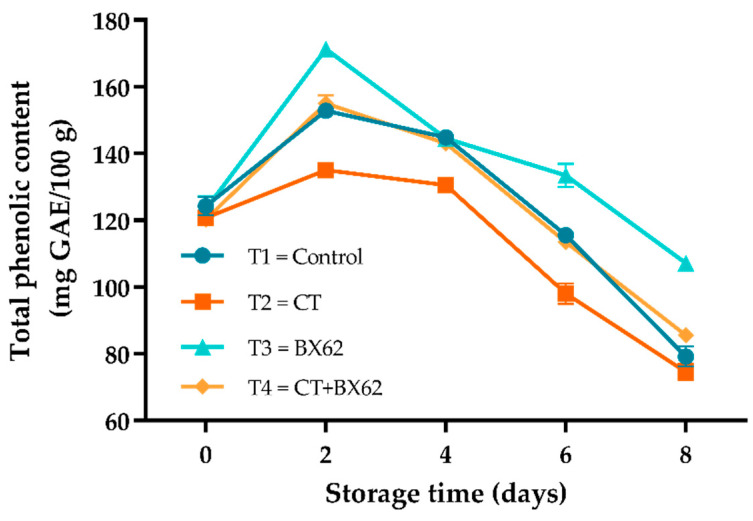
Changes in TPC for the control apples (●), chitosan-treated apples (■), *L. plantarum* BX62-treated apples (▲), and chitosan in combination with *L. plantarum* BX62-treated apples (◆) after being stored at 4 °C for 8 days. Each data point is the mean of three replicate samples.

**Figure 5 microorganisms-09-02404-f005:**
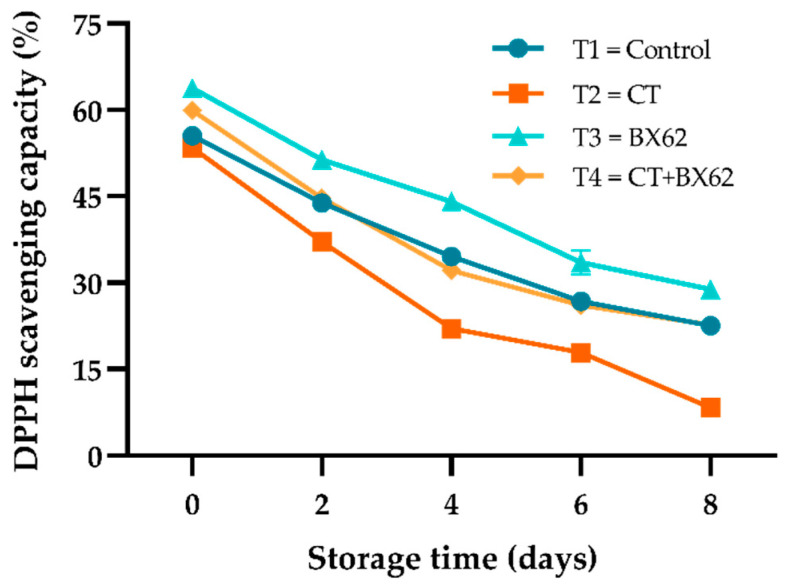
Changes in DPPH scavenging capacity for the control apples (●), chitosan-treated apples (■), *L. plantarum* BX62-treated apples (▲), and chitosan in combination with *L. plantarum* BX62-treated apples (◆) after being stored at 4 °C for 8 days. Each data point is the mean of three replicate samples.

**Figure 6 microorganisms-09-02404-f006:**
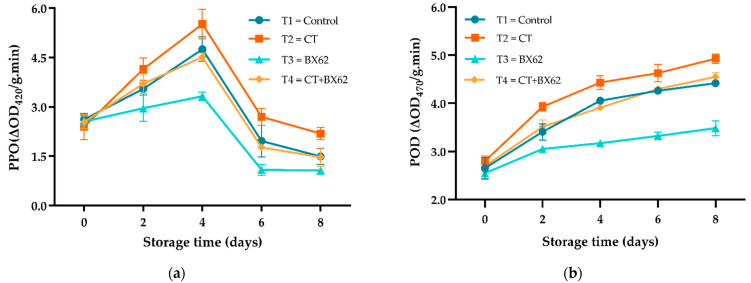
The changes of PPO (**a**) and POD (**b**) activities for the control apples (●), chitosan-treated apples (■), *L. plantarum* BX62-treated apples (▲), and chitosan in combination with *L. plantarum* BX62-treated apples (◆) after being stored at 4 °C for 8 days. Each data point is the mean of three replicate samples.

**Figure 7 microorganisms-09-02404-f007:**
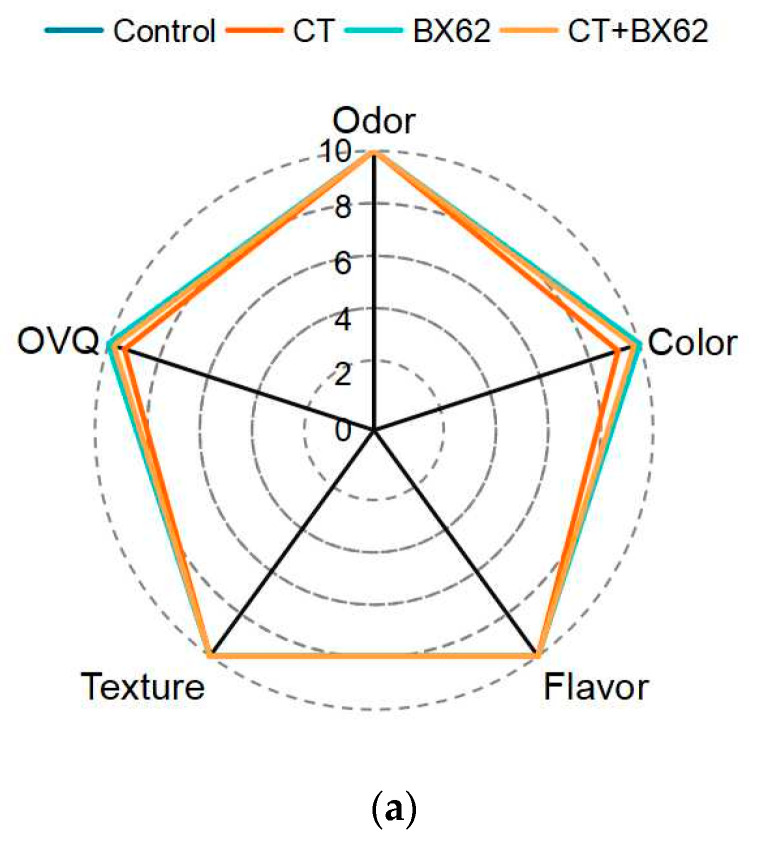
Sensory characteristics of fresh-cut apples treated with *L. plantarum* BX62 alone or incorporated with chitosan on days 0 (**a**), 2 (**b**), 4 (**c**), 6 (**d**), 8 (**e**).

**Table 1 microorganisms-09-02404-t001:** Changes in aerobic mesophilic bacteria (AMB), lactic acid bacteria (LAB), aerobic psychrophiles bacteria (APB), and yeast and molds (YAMs) with different treatments on fresh-cut apples during storage.

Microorganism	Treatment	Storage Time (Days)				
		0	2	4	6	8
AMB	Control	2.63 ± 0.07 ^b^	2.25 ± 0.08 ^ab^	2.53 ± 0.23 ^a^	2.61 ± 0.17 ^b^	3.31 ± 0.05 ^a^
	CT	2.68 ± 0.14 ^b^	<1.00 ^c^	<1.00 ^c^	<1.00 ^c^	<1.00 ^c^
	BX62	3.28 ± 0.11 ^a^	2.52 ± 0.21 ^a^	2.56 ± 0.22 ^a^	2.49 ± 0.07 ^a^	2.29 ± 0.11 ^b^
	CT + BX62	3.11 ± 0.08 ^a^	2.05 ± 0.20 ^b^	1.75 ± 0.09 ^b^	1.19 ± 0.07 ^b^	<1.00 ^c^
LAB	BX62	8.98 ± 0.10 ^a^	6.13 ± 0.13 ^a^	3.53 ± 0.13 ^a^	3.33 ± 0.09 ^a^	2.53 ± 0.32 ^a^
	CT + BX62	8.94 ± 0.11 ^a^	5.15 ± 0.07 ^b^	2.84 ± 0.11 ^b^	2.11 ± 0.05 ^b^	0.00 ± 0.00 ^b^
APB	Control	1.63 ± 0.01 ^a^	2.45 ± 0.21 ^b^	2.78 ± 0.14 ^a^	3.21 ± 0.08 ^a^	3.55 ± 0.12 ^a^
	CT	1.54 ± 0.07 ^a^	<1.00 ^a^	<1.00 ^c^	<1.00 ^c^	<1.00 ^b^
	BX62	1.63 ± 0.07 ^a^	2.46 ± 0.11 ^a^	2.42 ± 0.05 ^b^	2.91 ± 0.18 ^b^	3.40 ± 0.14 ^a^
	CT + BX62	1.73 ± 0.05 ^a^	<1.00 ^b^	<1.00 ^c^	<1.00 ^c^	<1.00 ^b^
YAMs	Control	1.63 ± 0.01 ^a^	1.87 ± 0.09 ^a^	2.31 ± 0.06 ^a^	2.65 ± 0.05 ^a^	2.58 ± 0.03 ^a^
	CT	1.52 ± 0.06 ^a^	<1.00 ^c^	<1.00 ^b^	<1.00 ^b^	<1.00 ^b^
	BX62	1.57 ± 0.04 ^a^	1.54 ± 0.05 ^b^	2.28 ± 0.13 ^a^	2.36 ± 0.10 ^a^	2.54 ± 0.13 ^a^
	CT + BX62	1.60 ± 0.11 ^a^	<1.00 ^c^	<1.00 ^b^	<1.00 ^b^	<1.00 ^b^

^a–c^ Different lowercase superscripts in the same column indicate significant differences (*p* < 0.05). CT = chitosan; BX62 = *Lactiplantibacillus plantarum* BX62.

**Table 2 microorganisms-09-02404-t002:** Correlation between some selected quality parameters of fresh-cut apples in different groups during cold storage.

Index	Pearson Correlation Coefficient (r)			
	Control	CT	BX62	CT + BX62
WLR vs. AMB	0.588	−0.875	−0.986 **	−0.939 *
WLR vs. APB	0.998 **	−0.875	0.922 *	−0.904 *
WLR vs. YAMs	0.930 *	−0.875	0.765	−0.904 *
WLR vs. LAB	--	--	−0.956 *	−0.991 **
BR vs. AMB	0.633	−0.659	−0.743	−0.904 *
BR vs. APB	0.978 **	−0.659	0.842	−0.687
BR vs. YAMs	0.979 **	−0.659	0.973 **	−0.687
BR vs. LAB	--	--	−0.924 *	−0.953 *
SSC vs. AMB	−0.460	0.895 *	0.953 *	0.935 *
SSC vs. APB	−0.985 **	0.895 *	−0.951 *	0.867
SSC vs. YAMs	−0.927 *	0.895 *	−0.855	0.867
SSC vs. LAB	--	--	0.982 **	0.990 **
DPPH vs. AMB	−0.573	0.810	0.879 *	0.927 *
DPPH vs. APB	−0.994 **	0.810	−0.967 **	0.835
DPPH vs. YAMs	−0.977 **	0.810	−0.911 *	0.835
DPPH vs. LAB	--	--	0.956 *	0.988 **
POD vs. AMB	0.475	−0.900 *	−0.962 **	−0.955 *
POD vs. APB	0.979 **	−0.900 *	0.971 **	−0.840
POD vs. YAMs	0.967 **	−0.900 *	0.842	−0.840
POD vs. LAB	--	--	−0.971 **	−0.992 **

****** The value was significant at *p* < 0.01, ***** The value was significant at *p* < 0.05. WLR = weight loss rate; BR = browning rate; SSC = soluble solids content; POD = peroxidase activity.

## Data Availability

The data presented in this study are available on request from the corresponding author. The data are not publicly available due to confidentiality.

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
