# Peer review of "The Effect of Lactiplantibacillus plantarum BX62 Alone or in Combination with Chitosan on the Qualitative Characteristics of Fresh-Cut Apples during Cold Storage"

_microorganisms, 2021, doi:10.3390/microorganisms9112404_

Round 1

Reviewer 1 Report

GENERAL COMMENTS

The present study aims to evaluate the effects of Lactobacillus plantarumBX62 (recently named Lactiplantibacillusplantarum) alone or in combination with chitosan, on the qualitative characteristics of fresh-cut apples during cold storage.

The paper should be revised by a native or fluent English speaker.

I have sincerely major reservations about the paper for numerous shortcomings.

In order to improve the clarity and the completeness of the work numerous modifications should be taken into account and so I think that this paper is not acceptable for publication in this form.In my opinion the biggest weakness of the work is the lack of evaluation of the sensory characteristics of the new products obtained.This aspect is very important in fresh ready-to-eat products.

TITLE

There has been extensive taxonomic restructuring in the family Lactobacillaceae so I suggest the authors rename Lactobacillus plantarum to Lactiplantibacillusplantarum (see doi https://doi.org/10.1099/ijsem.0.004107) in the title and throughout the manuscript.

I also suggest that the authors change the manuscript title. In the paper, the effect of Lactiplantibacillusplantarum BX62, alone or in combination with chitosan, on the qualitative characteristics of fresh-cut apples during cold storage was studied.

INTRODUCTION:

Line 56: I suggest to add other references related to the use of Lactobacillus plantarum as biopreservation agent for fruit and vegetables. I can suggest:

  • https://doi.org/10.1111/1750-3841.12171
  • https://doi.org/10.1016/j.fm.2014.11.008

MATERIALS AND METHODS

How was made fresh-cut apples coating with L. plantarum BX62?

Authors should be report the molecular weight of the chitosan used.

Why did you only use one percentage of chitosan?

Why microbial loads of Enterobacteriaceae and Faecal and Total coliforms were not evaluated? In my opinion these evaluations had to be done.

RESULTS

I suggest you check TITRATABLE Acidity % results (Figure 3a) of fresh-cut apple samples stored under different conditions.

I suggest the authors also check the microbiological results.

Apples are a good source of water and nutrients and are excellent substrates for microbial growth. How do the authors explain the absence of all microbial group investigated in chitosan-treated apples from day 2 of cold storage?Authors should indicate microbial loads <1 log cfu/g!!!

Why do LAB microbial loads of fresh-cut apples obtained with L. plantarum BX62 coating decrease during cold storage?

Author Response

Dear reviewer,

 We sincerely thank the reviewer for thoroughly reviewing our manuscript and providing very helpful comments to guide our revision, which have helped improve our manuscript. We have tried our best to revise the manuscript according to your kind and constructive comments and suggestions. Please find the following detailed responses to your comments and suggestions.

Reviewer 2 Report

Comments to the paper “Characterization of Lactobacillus plantarum BX62 Incorporated with Chitosan on the Quality of Fresh-cut Apples during Cold Storage” by Zhao et al.

General comments: the topic of the present work is particularly interesting, since viable cells are incorporated into a chitosan film to prolong the shelf life of fresh cut apples. The approach is also interesting because chitosan can be easily extracted from food wastes and by-products, providing a valorisation of these production wastes. However, I have several doubts about the work on the whole that need to be addressed.

The topic falls within AEM scopes and the manuscript is of interest for Microorganisms readers. I am not the best person to judge the English style, but the manuscript is clearly written and easy to follow.

Major points:

  • Why only one Lactiplantibacillus plantarum strains? How was it selected? Based on what? How can the authors be sure to have taken into account the best strain to validate the system? In work like this more strains are used for two main reasons: i) to discriminate about species (different strains of different species); ii) to validate the results at species level (more strains of the species showing the best results). Of course, I am not asking to perform other tests, but please discuss about this point. This is important for strains selection to drive future application of this strategy at industrial level.
  • Due to the target of the journal, it sounds odd that no microorganisms have been identified after plate counts. This should be of relevance for the discussion section. As is, the manuscript seems to fit better for another MDPI journal, like FOODS.
  • One of main questionable point of the strategy applied is the acidification capacity of the group of lactobacilli. The authors should check if the apples are acceptable by consumer. Thus, I strongly suggest a sensory evaluation of the final products.

Minor points:

  • Lactobacillus nomenclature is old and this is no more acceptable. Thus, in title and throughout the text, Lactobacillus genus nomenclature HAS TO be revised as reported by Zheng et al. (Int. J. Syst. Evol. Microbiol. 2020, 7, 2782-2858). In this case, Lactobacillus plantarum is actually Lactiplantibacillus plantarum.
  • The abstract is not descriptive enough. The main numerical results should be reported (in form of %, concentrations, activity, levels of microorganisms etc.). As is, this section, that should stand alone independently of the text, lacks of the more relevant data. E.g. since the system is applied to prolong shelf life, at least the decrease of the main microbial groups should be clearly reported.
  • “medium, The broth culture” something is wrong, please verify and correct.
  • How can the authors be sure that all hypochlorite was removed from apples? Was the residual hypochlorite concentration evaluated?
  • Antioxidant properties. Why did the authors determine the antioxidant capacity of apple pieces only the DPPH method? What about ABTS and FRAP methods?
  • Paragraph 2.8. The authors reported the reference of the method, what is the point in explaining it? Please be concise.
  • All figures are of very bad quality, please improve resolution.
  • L430-446. The entire section is scaring. The authors did not identify any bacteria, what is the point to discuss about this?

Author Response

(The authors gave the same response as above.)

Reviewer 3 Report

The manuscript entitled “Characterization of Lactobacillus plantarum BX62 Incorporated with Chitosan on the Quality of Fresh-cut Apples during Cold Storage” by Zhao et al. provides interesting information on the possibilities to incorporate a strain of Lactobacillus plantarum into a chitosan film in order to prolong the shelf life of fresh cut apples. The authors focused on the microbiological and physicochemical aspects of the apple slices during refrigerated storage. In my opinion the materials and methods are good described and the results are well explained and correlated with the literature but I have some comments, as follow:

- Authors should rename Lactiplantibacillus plantarum according to the new classification (http://lactotax.embl.de/wuyts/lactotax/).

- The abstract needs to be reordered, it does not stand alone the way it is actually written. There are no precisely defined research objectives. The abstract should include the purpose. Furthermore, the abstract should be reported the result in terms of percentages, levels, concentrations.

- The introduction section is sound but the main hypothesis it is not comprehensible and in my opinion they should be reported in a clear manner at the end of introduction.

- Line 104-105: Why was chosen this strain of L. plantarum? Was previously screened for the production of antimicrobial compounds?

- Line 127: it was necessary to analyze the sodium hypochlorite content that remained in the apples.

- Section 2.2. Please, include the number of replicates.

- Line 208-211: The authors should cite the literature to ensure the validity of the method.

- Line 211: add a dot after the round brackets.

- Section 2.9. Considering that the authors evaluate the quality of fresh-cut apples during cold storage in addition to yeast, molds, lactic acid bacteria and total mesophilic microorganisms, at least Enterobacteriaceae and Pseudomonadacee should have been investigated. However considering, that was used a selected strain the isolation and the application of polymorphic DNA profile recognition by Randomly Amplified Polymorphic DNA analysis is necessary to determinate the dominance of the added strains over indigenous microorganisms.

- Line 226: please delate a dot.

- Figures. Please improve the quality and add a statistical analyses.

- Tables 1: Please provide P values.

- Line 466. Is wrong. In this study it was not evaluated the ability of coating to reduce pathogen populations.

- A sensory evaluation is necessary in order to evaluate the overall satisfaction of the final product.

Author Response

(The authors gave the same response as above.)

Round 2

Reviewer 1 Report

Authors improved their paper and the current version is ready to accept.

Reviewer 2 Report

The authors addressed all comments, the text was modified accordingly or th erebuttal provided were convincing. I have no more doubts about the paper.

Reviewer 3 Report

The manuscript has been improved according to all reviewers suggestions and now I feel is ready for publication.